# Insulin Resistance and Hypertension: Mechanisms Involved and Modifying Factors for Effective Glucose Control

**DOI:** 10.3390/biomedicines11082271

**Published:** 2023-08-15

**Authors:** Hussein F. Sakr, Srinivasa Rao Sirasanagandla, Srijit Das, Abdulhadi I. Bima, Ayman Z. Elsamanoudy

**Affiliations:** 1Department of Physiology, College of Medicine and Health Sciences, Sultan Qaboos University, Muscat 123, Oman; 2Department of Human and Clinical Anatomy, College of Medicine and Health Sciences, Sultan Qaboos University, Muscat 123, Oman; srinivasa@squ.edu.om (S.R.S.); s.das@squ.edu.om (S.D.); 3Department of Clinical Biochemistry, Faculty of Medicine, King Abdulaziz University, Jeddah 21465, Saudi Arabia; hadibima@hotmail.com (A.I.B.); ayman.elsamanoudy@gmail.com (A.Z.E.)

**Keywords:** hypertension, insulin, resistance, diet, mechanism

## Abstract

Factors such as aging, an unhealthy lifestyle with decreased physical activity, snacking, a standard Western diet, and smoking contribute to raising blood pressure to a dangerous level, increasing the risk of coronary artery disease and heart failure. Atherosclerosis, or aging of the blood vessels, is a physiological process that has accelerated in the last decades by the overconsumption of carbohydrates as the primary sources of caloric intake, resulting in increased triglycerides and VLDL-cholesterol and insulin spikes. Classically, medications ranging from beta blockers to angiotensin II blockers and even calcium channel blockers were used alone or in combination with lifestyle modifications as management tools in modern medicine to control arterial blood pressure. However, it is not easy to control blood pressure or the associated complications. A low-carbohydrate, high-fat (LCHF) diet can reduce glucose and insulin spikes, improve insulin sensitivity, and lessen atherosclerosis risk factors. We reviewed articles describing the etiology of insulin resistance (IR) and its impact on arterial blood pressure from databases including PubMed, PubMed Central, and Google Scholar. We discuss how the LCHF diet is beneficial to maintaining arterial blood pressure at normal levels, slowing down the progression of atherosclerosis, and reducing the use of antihypertensive medications. The mechanisms involved in IR associated with hypertension are also highlighted.

## 1. Introduction

Hypertension is the most significant reversible risk factor for the development of cardiovascular disease (CVD) and is a cause of death all over the world [1,2]. In 2010, approximately 1.39 billion adults suffered from hypertension worldwide [3]. From 2000 to 2010, the age-standardized prevalence of hypertension was found to decrease by 2.6% in high-income nations compared to an increase by 7.7% in low-income and middle-income nations [3]. Due to the aging population, increased exposure to lifestyle risk factors such as poor diets (e.g. high salt and low potassium consumption), and a lack of physical activity, the incidence of hypertension is increasing worldwide [3]. However, there are regional variations in the incidence of hypertension. Hypertension has become a pandemic in recent decades, as the majority of the population is affected. 

Hypertension is mainly related to aging, environmental stress, and genetic susceptibility [4]. The prevalence of hypertension has changed, although not consistently, all over the world. High-income countries have shown a decline in the prevalence of hypertension during the last two decades, whereas low- and middle-income countries have witnessed a notable increase [3]. Since hypertension is responsible for causing various problems, the main aim of the present review was to highlight the role of insulin resistance (IR) in inducing hypertension and the detailed mechanism of its pathogenesis. We also discuss the dietary factors that cause IR and reveal that a low-carbohydrate, high-fat (LCHF) diet can be beneficial for reversing this condition. 

### Hypertension: The Silent Killer

As most hypertensive patients do not exhibit any symptoms, the disease is sometimes referred to as the “silent killer” and may be fatal due to cerebral strokes and ruptured arterial aneurysms. Heart disease, coronary artery disease, and stroke are just a few of the major issues that high blood pressure can cause. Therefore, it is critical to understand high blood pressure risk factors and take preventative measures [5]. Hypertension is defined as a systolic BP ≥ 140 mmHg and/or a diastolic BP ≥ 90 mmHg [3]. Primary hypertension is also essential hypertension which affects the population aged 35 years or older [6]. Secondary hypertension is common in teenagers or the elderly due to endocrinal disorders, which include acromegaly, Cushing’s syndrome, Conn’s syndrome and Grave’s disease, renal disorders (chronic renal failure and glomerulonephritis), and pregnancy (pre-eclampsia and eclampsia) [7]; at times, secondary hypertension may also be drug-induced. Modern medicine manages arterial hypertension through lifestyle modification, salt restriction, reducing caloric intake from saturated fats, and the use of antihypertensive medications [8]. Drugs used to lower arterial blood pressure include B-adrenergic blockers, ACE inhibitors, Angiotensin II receptor blockers, and vasodilators like calcium channel blockers. These medications are well known to cause side effects like bradycardia, heart block, hypertriglyceridemia, hypokalemia, dry cough, and postural hypotension [9]. 

The relationship between IR and arterial hypertension is highlighted in this review. A search of the literature was done by retrieving articles from databases such as PubMed, PubMed Central, and Google Scholar that were published during the period from 2000 to 2022. We did not perform the literature search as this was a systematic review. The main aim was to include facts and describe how insulin resistance induces systemic hypertension. 

## 2. Insulin Resistance 

IR is defined as the decreased ability of insulin to insert the glucose transporter (glut4) on the cell membrane facilitating glucose entry into body cells [10]. IR in the hepatocytes is associated with increased de novo lipogenesis and gluconeogenesis, leading to hypertriglyceridemia and hyperglycemia. De novo lipogenesis and increased VLDL output from the hepatocytes can lead to non-alcoholic fatty liver disease (NAFLD) and visceral obesity that exacerbates IR [11]. IR in the skeletal muscle fibers is associated with a decreased capacity to uptake blood glucose efficiently and predisposes post-prandial hyperglycemia [12]. Glucagon secreted by the alpha cells in the islets of Langerhans of the pancreas is responsible for glycogenolysis, gluconeogenesis, and amino acid metabolism [13]. Dysregulation of glucagon is also responsible for worsening diabetes, and it has been reported that in insulin-resistant α-cell lines, there is a disturbance in the regulation of glucagon secretion and expression [13]. Hence, IR is associated with a disturbance in glucagon secretion and expression.

Normal human adipose tissue lipolysis is inhibited by insulin; however, in people with IR, the process is sped up, increasing the release of FFA into the bloodstream [14]. The FFA are then deposited in other organs, such as the liver and muscle, leading to hepatic and skeletal muscle IR. Distorted adipokine production in adipocytes is also associated with IR. Resistin [15], leptin [16], TNF-a, and IL-6 [15,17] are a few adipokines that are increased, whereas others that improve insulin sensitivity are downregulated (e.g., adiponectin, visfatin, omentin) [18]. 

IR can be caused by multiple factors, including lifestyle, genetics, aging, and hormonal disturbances. Genetics affects the pathophysiology of IR and type 2 diabetes mellitus (T2DM). Children who had at least one parent with IR were shown to be more insulin-resistant themselves, with fasting insulin levels being around 20% higher than children who did not have any parents with IR [19]. By focusing on identical twins, researchers have examined the effect of familial genetics. As expected, there is a high possibility that individuals who share the same genes but were raised in different households would experience the same health issues, such as IR [20]. Importantly, only 5% of all occurrences of T2DM (and even fewer cases of prediabetes/IR) are caused by genetic abnormalities that result in IR [21]. It is interesting to note that some ethnic groups, which may contain a variety of distinct genetic features, are more likely to develop IR. One significant study evaluated the insulin sensitivity of four significant American racial groups: Hispanic, Asian, African, and “Caucasian” (which could more appropriately be characterized as “Northern European”) [22]. The group that was most likely to exhibit IR was Hispanic Americans, despite all groups having comparable body weights and waist-to-hip ratios. Asian Americans were the group with the second-highest levels of IR, which is surprising because they also had the lowest body weight and waist-to-hip ratio (although these differences were statistically insignificant). Caucasians had the lowest levels of IR, whereas African Americans had the third-highest level. It makes sense that obesity and the ratio of the waist to the hip are significantly linked to IR in the majority of the population. However, the Asian group did not appear to adhere to the same norms. This group was notably more likely to be IR despite having the lowest waist-to-hip ratios and BMIs [22]. 

Aging is another main culprit in IR. As people grow older, they become more IR [23]. Aging causes sarcopenia and muscle loss [24], leading to a decreased capacity to uptake blood glucose after meals and post-prandial hyperglycemia. Furthermore, hormonal changes that appear with aging, like lowered growth hormone and dehydroepiandrosterone (DHEA) levels [25], may also contribute to IR. Hormonal changes are involved in the pathogenesis of IR. From clinical observations, it is evident that testosterone declines in elderly men in a form of male menopause. Similar to how a woman’s body changes as a result of decreased estrogen, a man’s body changes as a result of decreased testosterone. Increased IR is one of these alterations [26]. One can lessen these adverse effects and raise insulin sensitivity by administering male testosterone [27]. Male hypogonadism in rats is associated with decreased insulin sensitivity and NFLD through increased expression of levels of SREBP-1, SREBP-2, ACC-1, FAS, HMGCOAR, and HMGCOAS with increased protein levels of both precursor and mature SREBP-1 and SREBP-2, PPAR-α, p-PPAR-α, CPT-1, and UCP-2 and significantly lower protein levels of p-AMPK and p-ACC-1 [28]. Normal androgen levels in males prevent the build-up of hepatic fat, but androgen insufficiency results in hepatic steatosis. Higher androgen levels in females can raise the risk of NAFLD in PCOS [29]. The sex hormone binding protein SHBG is involved in the control of androgen. SHBG has recently been suggested as a substitute marker for NAFLD [30]. Estrogen supports the maintenance of IR in both males and females. Individuals who are unable to produce estrogens owing to a deficit in aromatase (the enzyme that converts androgens to estrogens), provide substantial proof for this fact. Along with other consequences, the inability to manufacture estrogen leads to the development of IR [31]. Artificially maintaining high estrogen levels with hormone treatment helps preserve insulin sensitivity through menopause. Hence, estrogen deficiency is associated with IR [32]. 

An intriguing combination of neuronal and endocrine (hormone) activities is involved in the stress response. Both cortisol and epinephrine are released by the adrenal gland. Epinephrine raises blood pressure and heart rate during the initial phases of stress. However, prolonged exposure to excessive epinephrine may result in IR. In a previous study, healthy males received several evaluations to gauge their insulin sensitivity, either with or without an epinephrine infusion. A two-hour adrenaline injection reduced insulin sensitivity by more than 40% [33]. Long-term stress is associated with increased cortisol production from the adrenal cortex through the activation of the hypothalamo–hypophyseal–adrenal axis [34]. Cortisol exerts anti-stress effects through increased energy sources like glucose by gluconeogenesis, FFA by lipolysis, and circulating amino acids by protein catabolism. Increased levels of glucose and FFA stimulates insulin production and lipogenesis with the accumulation of fat around the viscera, thereby leading to visceral obesity. In addition, it increases the smooth muscle response to circulating catecholamines, increasing arterial blood pressure [35]. Another group of hormones that affect cellular metabolism are thyroid hormones, which are linked to IR, especially in hypothyroidism or thyroid resistance. TSH is correlated with obesity, and obesity results in IR [36]. There is an instance of “thyroid resistance,” in which the body responds to thyroid hormone control less effectively. Unexpectedly, reducing weight tends to lower thyroid levels, indicating that thyroid hormones are more potent and the body is more susceptible to their effects [37]. Importantly, hypothyroidism alters how fat cells react to insulin. Hypothyroidism reduces the amount of glucose that fat cells can take up in response to insulin, but insulin can still stop fat breakdown in the fat cell, limiting cell shrinkage. As a result, hypothyroidism inhibits fat loss as blood insulin levels rise [38].

The most significant independent predictor for the development of Type 2 Diabetes (T2DM) has been suggested to be IR. Obesity, abdominal adiposity, IR, and ultimately T2DM have all been linked to chronically consuming too many calories from foods like fructose, trans-fats, polyunsaturated fatty acids, and grains, as well as engaging in insufficient physical exercise [39].

## 3. Diet-Induced Insulin Resistance 

### 3.1. Fructose 

Along with the increase in daily calorie consumption that has occurred over the past three decades, the prevalence of obesity, diabetes, and IR has grown considerably [40]. Unmatched by physical activity or energy demand, the constant supply of energy from dietary carbohydrates, lipids, and protein fuels may lead to a backlog of mitochondrial oxidation products, which is a mechanism linked to increasing mitochondrial malfunction and IR [41].

High-fructose corn syrup (HFCS), which contains 42% or 55% fructose and glucose respectively, and sucrose, which includes 50% fructose and 50% glucose, are the two most common forms of fructose consumption [42]. The secular increases in fructose intake, in contrast to those of trans fats or ethanol, have coincided with the growth of obesity and metabolic syndrome (MetS), particularly in children. Before World War II, Americans consumed 24 g of fructose per day; by the middle of the 1970s, the amount had increased to 37 g; and by the middle of the 1990s, it had reached 55 g (representing a gradual increase from 5% to 7% to 10% of total calories) [43]. In fact, according to current National Health and Nutrition Examination Survey (NHANES) statistics, almost 15% of US citizens obtain 25% of their daily calories from added sugars [44]. Interestingly, excessive fructose ingestion has been linked to weight gain, visceral adiposity, dyslipidemia, IR/glucose intolerance, and NAFLD [11]. This is especially true of drinks with added sugar. Independent of adenosine triphosphate hydrolysis and salt absorption, fructose is delivered into the enterocyte in the small intestine via the fructose transporter Glut5 [45]. Nevertheless, most of the fructose that is consumed is absorbed into the portal circulation and transported to the liver. There, fructokinase quickly converts fructose to fructose-1-phosphate (F1P), a mechanism that is insulin-independent and sidesteps the glycolytic pathway’s negative feedback control of phosphofructokinase [45]. Thus, in an uncontrolled manner, the mitochondria are directly supplied with lipogenic substrates (such as glyceraldehyde-3-phosphate and acetyl-CoA) produced by fructose metabolism. Due to the overproduction of mitochondrial precursors, hepatic DNL can overpower apoB and the lipid export system, resulting in intrahepatic lipid accumulation and steatosis [46]. Hepatic DNL also restricts further fatty acid oxidation in the liver through excess production of malonyl-CoA, which reduces the entry of fatty acids into the mitochondria. F1P also activates SREBP-1c via peroxisome proliferator-activated receptor-g coactivator-1 independently of insulin, which activates the genes involved in DNL [47]. Furthermore, JNK-1, a liver enzyme that links hepatic metabolism and inflammation, is stimulated when F1P activates dual specificity mitogen-activated protein kinase [48]. Additionally, PKC is activated by the lipogenic intermediate diacylglycerol, which is created during the liver’s processing of fructose. Both of these occurrences promote IRS-1’s serine phosphorylation, which results in hepatic IR [49]. 

When fructose enters hepatocytes, fructokinase quickly phosphorylates it into fructose-1-phosphate. As part of this process, ATP contributes phosphate, creating ADP, which is then converted into uric acid [50]. A fructose-mediated rise in AMP deaminase facilitates this process [51]. Elevated uric acid levels may cause oxidative stress in hepatocyte mitochondria, enhancing the generation of ROS and eventually leading to mitochondrial dysfunction [52]. Uric acid stimulates hepatic gluconeogenesis through the activation of AMPD2 and AMPK blockade [53]. Uric acid accumulation leads to insulin resistance [53,54], decreased FFA oxidation, endoplasmic reticulum stress, and decreased NO generation, leading to elevated blood pressure [55]. The involvement of fructose in the pathogenesis of IR is depicted in Figure 1.

### 3.2. Omega-6 Fatty Acids

Vegetable oils like canola, sunflower, corn, soy, and safflower oils are rich sources of omega-6 fatty acids, especially linoleic acid [56,57]. Since 1900, vegetable oils derived from seeds and flowers have been extensively used to replace saturated fats such as butter, tallow, and lard. The use of these oils gives processed food more stability. Linoleic acid, the primary omega-6 polyunsaturated fat found in vegetable oils like seed oils [57], increased by more than a factor of two as a result, and it currently accounts for 8% to 10% of the total energy consumed in the West [56]. Conjugated linoleic acid, which is present in pastured animal diets, should not be mistaken for the omega-6 fatty acid linoleic acid. In addition, because linoleic acid has a half-life of approximately two years in adipose tissue, the content of linoleic acid in adipose tissue is a trustworthy indicator of ingestion [58]. Oxidative stress, damage to the tissues, and mitochondrial dysfunction resulting from excess linoleic acid are responsible for various cardiovascular diseases, including Alzheimer’s disease, cancer, dementia, obesity, and diabetes [59]. Excessive consumption of linoleic acid in recent years has led to a reduction in the levels of omega-3 fatty acid intake, with an increasing ratio between omega 6 and omega 3 from 4:1 to 25:1 [60]. Significantly, as alpha-linolenic acid interacts with linoleic acid for conversion to longer-chain polyunsaturated fats, greater ingestion of omega-6 polyunsaturated fat linoleic acid can decrease omega-3 in the body [61]. Since it has long been known that individuals with CAD had lower levels of linoleic acid relative to total fatty acids in lipids, this finding has been used to support the theory that linoleic acid deficiency may be a risk factor for heart disease [62]. However, linoleic acid levels in both adipose tissue and platelets are favorably correlated with coronary artery disease (CAD), but long-chain omega-3 levels in platelets (eicosapentaenoic acid (EPA) and docosahexaenoic acid (DHA)) are negatively correlated with CAD [63]. Linoleic acid produces mitochondrial toxicity and catastrophic cardiolipin modifications [64], which are predisposed to exaggerate oxidative stress and overproduction of reactive oxygen species (ROS). ROS consume natural antioxidants and also produce lipid peroxidation of the cell membrane and DNA damage. Oxidative stress predisposes to IR, obesity, T2DM, heart failure, and cancer. A schematic diagram (Figure 2) shows how linoleic acid may cause inflammation and oxidative stress conditions. 

### 3.3. Grains 

Beginning as far back as ten thousand years ago, the use of wheat and other grains was linked to increased infections, bone illnesses such as osteoporosis, diabetes, metabolic syndrome, increased newborn mortality, and a shortening of life span [65]. Grains affect human health in different ways according to contents including amylopectin A and gluten protein. Therefore, we discuss in detail how grains predispose to IR and T2DM. The two most frequently produced and consumed grains in the world, wheat and maize, have undergone significant transformation over the past 50 years due to agribusiness [66]. 

While gluten is sometimes held responsible for all of the issues associated with wheat, in reality, gliadin, a smaller protein found inside gluten, is to blame for a number of the negative effects of contemporary wheat on human health. Gliadin proteins come in over 200 different varieties, many of which are weakly and imperfectly digested by humans and are therefore compatible with being found in grass seeds. The most potent cause of celiac disease, which is intestinal damage of the small intestine caused by wheat, rye, and barley, is a gene for a kind of gliadin called Glia-9 [67]. Only a few strains of wheat from the early 20th century included the Glia-9 gene, but it is now found in the majority of contemporary cultivars [68]. Many of the proteins found in grains are either not degraded at all or are broken down into minute fragments or peptides (short sequences of amino acids) rather than single amino acids. New gliadin variants are partially digested to produce small peptides that then enter the bloodstream and bind to opiate receptors in the neural network. These receptors are stimulated by heroin and morphine and have effects such as “mind fog”, paranoia, anxiety, the mania of bipolar disorder, depression, and appetite stimulation [69]. People who battle an insatiable hunger, such as those with bulimia and binge-eating disorders, are examples of the final impact of appetite stimulation, which is frequent and can be overpowering. These peptides are known as exorphins, which are exogenous morphine-like substances. However, gliadin-derived peptides do not provide a “high”, but rather an increase in appetite and caloric intake, with studies showing steady increases of 400 calories per day, largely from carbohydrates and grains [70]. Amylopectin is a complex carbohydrate composed of 25% amylose, a linear chain of glucose units, and 75% amylose, a chain of branching glucose units. Both amylopectin and amylose can be broken down by pancreatic and salivary amylase [71]. While part of the amylose is not completely digested, some of it goes to the colon undigested when the amylase enzymes hydrolyze amylopectin into glucose monomers [72]. Because it is most effectively digested, the complex carbohydrate amylopectin is immediately converted to glucose and transported into the circulation, resulting in an abrupt rise in blood glucose in response to the consumption of grains. According to a previous study, non-diabetic people with hyperglycemia are considerably more likely to experience brain shrinkage. IR is induced by hyperglycemia, which also causes hyperinsulinemia [73]. 

Gluten, a large protein made up of gliadin and glutenins, gives wheat dough its distinctive stretchiness. Since the long-branching glutenin proteins affect baking properties, gluten has undergone genetic manipulation [74]. Therefore, using unexpected breeding techniques, geneticists have crossed different wheat strains, mated wheat with non-wheat grasses to introduce novel genes, and employed chemicals and radiation to cause mutations in the glutenin component of gluten. Up to 14 distinct glutenin proteins that have never been consumed by humans are produced when two different wheat plants are hybridized [75]. As a result, novel glutenin protein genes not present in previous strains of wheat have been discovered in current strains of wheat; none of these genes, naturally, have been evaluated for acceptability for human consumption [76]. 

Wheat germ agglutinin (WGA), a protein found in wheat (as well as barley, rye, and rice) that defends the plant against molds and insects, has undergone structural alterations as a result of the genetic modifications imposed on wheat [77]. Due to agribusiness manipulation, the structure of modern wheat’s WGA varies from that of ancient wheat strains [78]. Ironically, geneticists have attempted to raise the amount of WGA in grains to boost pest resistance, which has increased the toxicity of such grains for human gastrointestinal systems. No matter how it is prepared (i.e., cooking, baking, sprouting the seeds, sourdough fermentation), people consume intact WGA, which belongs to the lectin protein family and is extraordinarily tough, entirely indigestible, and resistant to any breakdown by the body. WGA lectins that are not digested are hazardous because they cause harm without the necessity for hereditary predisposition [79]. Even in the absence of gliadin/gluten, WGA alone is sufficient to cause intestinal damage similar to celiac disease by altering microvilli, the absorptive “hairs” of intestinal cells [80]. 

Grain breeding efforts over the past 50 years have favored strains with higher phytate contents because phytic acid, or phytates, like WGA, protect the plant from pests. Additionally, since phytate content and fiber content are inversely correlated, recommendations to increase dietary fiber by consuming more “healthy whole grains” also result in a rise in phytate content. For instance, modern wheat, corn, and millet have the equivalent phytate levels of 800 mg/100 gm (312 ounces) of flour [81]. Iron, zinc, calcium, and magnesium absorption are all stopped by phytates at doses as low as 50 mg [82]. While passionate grain-consuming cultures can consume up to 5000 mg of phytates/per day, amounts that are related to nutritional deficits like zinc deficiency, osteoporosis, and iron deficiency anemia, children normally consume 600 to 1900 mg of phytates per day [80].

One of the most prevalent allergens is a group of proteins known as alpha-amylase inhibitors, which can lead to hives, asthma, cramps, diarrhea, and eczema [83]. Modern alpha-amylase inhibitors differ from conventional strains structurally by 10%, which can correspond to up to a dozen amino acid changes. Just a few amino acids can make the difference between the absence of an allergic reaction and a severe allergic reaction, even anaphylaxis. Baker’s asthma is a disorder that often strikes those who work in the baking business [83]. Additionally, there is a strange syndrome known as wheat-dependent exercise-induced anaphylaxis, which is a severe and sometimes fatal allergy brought on by physical activity after consuming wheat. Gliadin protein allergy is the cause of both diseases [84]. The schematic diagram in Figure 3 depicts the mechanism by which grains are involved with IR.

## 4. The Detailed Mechanism of Insulin Resistance Linked to Hypertension

IR is a well-known etiology of primary hypertension in adults, and several underlying mechanisms are involved. These mechanisms (discussed in detail below) increase the total peripheral resistance and the preload in the cardiovascular system [85].

### 4.1. Advanced Glycation End Products (AGEs) and Hypertension 

Glycation is the process of incorporating carbohydrates into a protein, lipid, or DNA. This process can happen enzymatically or non-enzymatically. Enzymatic glycation, such as creating a glycosidic bond while using a sugar donor during glycoprotein synthesis, is commonly referred to as glycation [86]. The non-enzymatic chemical interactions of reducing sugars with proteins are referred to as glycation reactions [86]. For instance, the reaction of glucose monomer with lysine residues in proteins results in the formation of a ketamine (Amadori) adduct. Terms like “glycation”, “fructation”, “ribation”, etc. are used to describe glycation caused by certain sugars [87]. Protein function is lost due to glycation, and tissues, including blood vessels, skin, and tendons, lose elasticity [88]. When there is hyperglycemia and tissue oxidative stress, the glycation process is greatly accelerated [89]. This suggests that glycation may have a role in developing diabetes complications and aging. The glycation process fits well with the hypothesis that the build-up of metabolic waste increases with aging since there are no enzymes to eliminate glycated compounds from the human body [90]. All of the body’s proteins, including circulating, extracellular, and intracellular proteins, are affected by glycation. Hemoglobin, albumin, insulin, immunoglobulins, low-density lipoproteins (LDL), and collagen are a few examples. Albumin, insulin, hemoglobin, immunoglobulins, lipoproteins, and fibrinogen are some of the proteins that come into direct contact with blood and, consequently, blood glucose. Glycation also targets other compounds containing amino groups, such as DNA. A sizable amount of research linking AGEs to atherosclerosis comes from studies on diabetes, an insulin-resistant illness linked to a high prevalence of vascular problems. Diabetes patients and diabetic animal models have higher levels of AGEs, which are linked to atherosclerotic alterations as seen in various studies [91]. Glycation plays a role in IR’s biological processes. Furthermore, glycated insulin in pancreatic cells or the blood makes it more challenging to maintain glucose homeostasis and induce lipogenesis [92]. Finally, glycated albumin and methylglyoxal (a glycation mediator) has been demonstrated in various investigations to reduce the physiologic reactions generated when insulin binds to its receptor on the target cell. AGEs are related to a rise in vascular stiffness in uncontrolled essential hypertensives [93]. Cardiovascular risk is enhanced by increased vascular stiffness and systolic hypertension as a result [94]. Although this syndrome may be worsened by diabetes or other disorders in which AGE generation is enhanced, it is frequently encountered with aging [95]. AGEs cause hypertension through increased oxidative stress. The endogenous antioxidants cysteine and glutathione are depleted at high concentrations of methylglyoxal, which reduces their antioxidant ability. Additionally, glutathione reductase and glutathione peroxidase, antioxidant enzymes containing sulfhydryl and amino groups at their active sites, are inhibited by methylglyoxal and AGEs, which reduces their capacity to break down ROS [96]. This was shown to occur in cultured rat vascular smooth muscle cells, in which the addition of methylglyoxal blocked these enzymes, resulting in oxidative stress, low levels of reduced glutathione, and higher amounts of oxidized glutathione [97]. When compared to normotensive rats, spontaneously hypertensive animals had lower levels of glutathione, tissue glutathione peroxidase, and reductase activities and higher levels of plasma methylglyoxal, aortic AGEs, and oxidative stress [98].

Additionally, ACEs induce endothelial dysfunction and reduce NO synthesis by the endothelial cells, causing vasoconstriction and increasing total peripheral resistance. NO is produced from the amino acid arginine by the enzyme nitric oxide synthase (eNOS) and the co-factor tetrahydrobiopterin (BH4). The generation of NO is partly controlled by insulin, which interacts with certain cell surface receptors [99]. By generating AGEs with arginine, excess methylglyoxal and other aldehydes may restrict substrate [100].

Additionally, the AGEs increase the activity of the renin–angiotensin–aldosterone system (RAAS) [101]. The interaction between AGEs, the AGE-RAGE system, and the RAAS system is becoming more and more clear. When administered to Sprague Dawley rats, AGE-modified rat serum albumin enhanced renal expression of ACE, renin, angiotensinogen, and angiotensin receptor type 1, increased renal ACE activity, and was linked to glomerular and tubular hypertrophy as well as AGE buildup. A type 1 (AT1R) angiotensin receptor blocker counteracted these effects [66]. The administration of AII to these animals caused renal hypertrophy and salt retention as well as an increase in the serum and renal accumulation of AGEs and advanced oxidation protein products. These modifications were decreased by the AGE inhibitor pyridoxamine [66]. The LCHF diet stabilizes blood glucose levels and reduces spikes after meals, consequently lowering the synthesis of AGEs. Restricting CHO intake in patients with IR and T2DM to <30% of daily calories improves the level of HbA1C, one of the glycation-derived AGEs [102]. In another study, CHO restriction was found to reduce AGEs in the kidney [103].

### 4.2. Oxidative Stress and Hypertension

Following a disturbance in the synthesis and accumulation of reactive oxygen species (ROS) in cells and tissues, the ability of a biological system to detoxify these reactive products causes oxidative stress, even though ROS are generally produced as byproducts of oxygen metabolism and can play various physiological roles. Environmental stressors (e.g., UV, ionizing radiation, pollutants, and heavy metals) and xenobiotics (e.g., antiblastic drugs) contribute to significantly increased ROS production, causing the imbalance that leads to cell and tissue damage (oxidative stress) [104]. ROS include superoxide radicals (O_2_•^−^), hydrogen peroxide (H_2_O_2_), hydroxyl radicals (•OH), and singlet oxygen (^1^O_2_), which are produced as physiological byproducts in biological systems [69,105].

Fibrosis, perivascular inflammation, and vascular calcification are important characteristics of hypertension that are crucial in vascular aging and contribute to vascular stiffness [67]. Vascular smooth muscle cell (VSMC) adhesion and stiffness are both increased by hypertension, and these effects increase with age [68]. 

Collagen and elastin, two important extracellular matrix (ECM) scaffolding proteins that are responsible for the structural strength and elasticity of the arteries, respectively, are essential for vessel wall compliance [106]. “Adaptive” vascular remodeling enables the arteries to adjust to momentary hemodynamic changes in the early stages of hypertension [107]. Although arterial compliance is decreased functionally, persistent and severe increases in blood pressure can cause “hypertrophic maladaptive” remodeling of the arterial wall [108], which is histologically characterized by an increase in the media-to-lumen ratio, cross-sectional area, and arterial wall thickness [108]. Membrane type-1(MT1)- matrix metalloproteinase (MMP) and ROS both stimulate MMP-2. MMP-2 breaks down a variety of ECM and non-ECM components after activation [108]. Collagen and elastin are destroyed by MMP-2 at the extracellular level. The collagen’s cleaved byproducts attach to the VSMCs’ integrin receptors and activate focal adhesion kinase, which promotes cell migration and proliferation. MMP-2 further supports the VSMC transition to more proliferative, hypertrophic, and pro-inflammatory phenotypes by activating the latent form of TGF-alpha [109]. Activation of MMP-2 by ROS results in the proteolysis of calponin-1 at the intracellular level, supporting VSMC migration and proliferation. MMPs may cleave VEGFR-2 and eNOS in endothelial cells, resulting in vasoconstriction and capillary rarefaction. Antioxidants may reduce vascular remodeling brought on by hypertension by interfering with these pathways [110].

A study on rats showed that LCHF diet enhanced antioxidant defense, but insufficient evidence links the outcomes to mitohormesis [111]. Initially, the LCHF diet promotes the generation of small amounts of ROS, which subsequently activates the nuclear factor erythroid-derived 2 (NF-E2)-related factor 2 (Nrf2) antioxidant pathway, the principle stimulator of the detoxification genes [112,113,114]. The Nrf2 pathway was also upregulated by the administration of B-hydroxy butyrate [115].

### 4.3. Systemic Inflammation and Hypertension 

Visceral adipocytes are well-known to produce adipocytokines such as TNF-a, IL-6, and hs-CRP. Several studies have concluded that IR and Met S are associated with chronic inflammation processes [116]. Numerous cell types and released substances are involved in the complicated process of inflammation, which has been linked to high blood pressure in many instances. In animal models of essential hypertension and dendritic cells from hypertensive patients, ROS generation is increased [117] and this produces activation of T-lymphocytes. T-cells directly contribute to endothelial cell dysfunction and renal damage since they have been demonstrated to infiltrate the kidney and the vascular tree in animal models of essential hypertension [118]. It is thought that CD8+ “killer” T-cells in particular are responsible for direct cellular damage, which might be an autoimmune cause of hypertension. T-cell activation and function-related factors are crucial mediators of critical hypertension [118]. T-cells then travel to the kidney and vascular tree, where they cause hypertension and inflammation. Chronic inflammatory processes in the renal tissue are associated with glomerulosclerosis [119], RAAS activation [120], tissue damage, and vascular fibrosis. 

Inflammation of the vessel wall and the disturbance of the endothelial function share a role in the pathogenesis of atherosclerosis [109] beginning with the glycocalyx, which grows like hairs on the outer surface of the endothelial cells and manages how LDL-cholesterol moves to the arterial wall [121]. The glycocalyx can be damaged by ischemia-reperfusion injury, hypertension, oxidative stress, and hyperglycemia [122,123]. Nitric oxide (NO), an anti-atherogenic agent, is released by the underlying endothelial cells as a result of the shear stress caused by blood flow in arteries at the glycocalyx [124]. Due to a lack of NO production through this process, areas of the arterial tree with low shear stress are more prone to atheroma, whereas exercise is protective by increasing blood flow and shear stress [125]. Glycocalyx dysfunction due to luminal hyperglycemia has been demonstrated; thus, in individuals with T2DM and metabolic syndrome, this is the initial stage of the atherothrombotic process (IR). Additionally, there is proof that exposure to oxidized low-density lipoprotein causes glycocalyx problems [124]. LDL-cholesterol moves to the intima either via the trans-cellular or para-cellular route [126]. Para-cellular transport represents the leakiness of LDL-cholesterol through the tight junction [127]. Highly sensitive CRP, oxidized LDL-cholesterol, ROS, TNF-a, AII, and IL-17 destroy the endothelial cells, increasing the deposition of LDL-cholesterol into the wall of blood vessels and the formation of atheroma. Proteoglycans can trap LDL-cholesterol to undergo more oxidation and glycation reactions [128]. With hyperglycemia in IR, more proteoglycan can bind to LDL-cholesterol in diabetic rats compared to controls [129]. On the other hand, HDL-cholesterol, which is capable of removing the cholesterol from the vessel wall to the liver, is reduced in MetS and IR predisposing to atherothrombosis [130].

The effect of a LCHF diet on decreasing inflammation has been confirmed in multiple human and animal studies. The ketone bodies formed in response to the LCHF diet exert an anti-inflammatory effect by stimulating the synthesis of anti-inflammatory cytokines and lowering pro-inflammatory ones [131,132]. 

### 4.4. Dyslipidemia and Hypertension

In response to the standard Western diet rich in grains, high fructose corn syrup, and vegetable oils, there is increased formation of VLDL cholesterol in the hepatocytes. VLDL-cholesterol ends in the formation of small LDL-cholesterol particles that persist in the circulation for more than two weeks due to the inability of the LDL-cholesterol on the hepatocytes to uptake them. The sLDL-cholesterol is exposed to oxidative stress and glycation by hyperglycemia, increasing their uptake by the macrophages in the vessel wall and atherosclerosis [133]. Within the blood flow and vascular wall, the LDL particles go through several alterations that change their size, density, and chemical characteristics [134]. Oxidized LDL (ox-LDL) is created when LDL is oxidized by free radicals and IR [135]. Atherosclerosis progresses to oxidized LDL causing endothelial dysfunction and the gene production of adhesion molecules on the cell surface [134]. Additionally, Sierra-Johnson discovered that in the non-diabetic US population, a higher apo-B/apo-AI ratio is substantially linked to IR [136]. Theoretically, the apo-B100/apo-AI ratio is a more accurate predictor of CHD risk than traditional markers [137]. In IR and MetS, a rise in the triglyceride/high-density lipoprotein-cholesterol ratio (TG/HDL-C) is indicative of IR and dyslipidemia. According to a survey, 48% of people with metabolic syndrome had a TG/HDL-c ratio that was unusually high [138]. As a result, it is regarded as a MetS replacement marker [139] and a reliable sign of IR [140]. 

In response to the LCHF diet, several studies have investigated atherogenic parameters either in human or animal studies. LCHF modulates the TG/HDL cholesterol ratio [141] by reducing circulating TG and increasing HDL cholesterol. In another study, the LCHF diet increased the molecular expression of the hepatic VLDL receptor, which inhibits the release of triglycerides from the liver caused by VLDL and reduces the amount of LDL that is converted to triglycerides. Additionally, the LCHF diet boosts the activity of the lipoprotein lipase enzyme, which is responsible for hepatic TG lipolysis [142]. 

### 4.5. Endothelial Dysfunction and Hypertension

It is crucial to maintain a healthy balance between blood clot formation and clot lysis to keep the cardiovascular system functioning properly and provide continuous blood flow [143]. The final stage of homeostasis, known as fibrinolysis, removes extra fibrin from the circulatory system. The breakdown of fibrin occurs in two main phases. Plasminogen is first converted to plasmin by several crucial proteins and enzymes, including tissue plasminogen activator (t-PA) and urokinase plasminogen activator (u-PA) [144]. The active plasmin that may combine with fibrin selectively breaks down the complexed fibrin into soluble fibrin degradation products in the second and final stages [144]. Endothelial dysfunction is also characterized by one or more of the following characteristics i.e., decreased endothelium-mediated vasorelaxation, hemodynamic deregulation, impaired fibrinolytic ability, increased turnover, excessive growth factor creation, elevated expression of adhesion molecules and inflammatory genes, higher oxidative stress, and increased permeability of the cell layer [145]. Endothelial dysfunction may be involved in the development and progression of hypertension through several processes, including increased constriction and vascular remodeling (i.e., structural, mechanical, and functional modifications) of resistant arteries [146]. 

The activation of NO synthase (eNOS) and an increase in NO production are both caused by increased laminar shear stress, which simultaneously caused an increase in endothelial cytosolic calcium concentration [147]. Increased cytosolic calcium levels also cause calcium-activated potassium channels to open, which is linked to endothelial cells being hyperpolarized and, as a result, to vasorelaxation [148]. Additionally, a mechanosensory complex made up of caveolin, tyrosine-specific phospho-transferase Fyn, vascular endothelial growth factor (VEGF)-receptor 2, and platelet endothelial cell adhesion molecule-1 (PECAM-1) allows endothelial cells to respond adequately to the positive effects of shear stress [149]. On the other hand, oscillatory flow increases the secretion of pro-inflammatory molecules like MCP-1 (monocyte chemotactic protein 1), PDGFs (platelet-derived growth factor), and endothelin-1, which causes vasoconstriction, elevated blood pressure (BP), and the development of atherosclerosis in larger arteries [150]. Several transcription factors, including Kruppel-like factor [KLF2/4], NF-B, AP-1, early growth response-1, c-Jun, c-fos, and c-myc, as well as mitogen-activated protein kinases (MAPKs) and small ubiquitin-like modifier (SUMO) signaling, are activated during these processes, which also cause an increase in reactive oxygen species (ROS) [151].

### 4.6. Hypervolemia and Hypertension

Insulin has specific receptors on the renal tubules that increase Na reabsorption as angiotensin II (AII). Adipocytes also secrete angiotensinogen, which can be converted to AI. ACEs transform AI into AII, and consequently, AII stimulates the production of aldosterone hormone by the zona glomerulosa of the adrenal cortex. RAAS activation increases arterial blood pressure through elevated total peripheral resistance and blood volume. Insulin affects the whole nephron, as has long been understood. Using the radioisotope method, Bourdeau et al. demonstrated that insulin is deposited in the proximal tubule [152]. The rabbit nephron’s thick ascending limb of Henle’s loop and distal convoluted tubule is where insulin binds most strongly, as demonstrated by Nakamura et al. [153]. Butlen et al. demonstrated that insulin accumulates most heavily in the proximal tubule of the rat nephron, followed by the pars recta and distal convoluted tubule [154]. Insulin directly stimulates the Na+-H+ exchanger type 3 (NHE3) activity in the proximal tubules of rats, according to Gesek and Schoolwerth’s research [155]. This is significant because NHE3 is crucial for proximal tubules’ apical sodium entry. Akt is well recognized to play a crucial role in the phosphoinositide 3-kinase (PI3K)-mediated translocation of NHE3 into the apical membranes of proximal tubular cells, even if the signaling route of insulin-mediated NHE3 activation is still unknown [155,156,157]. The regulation of NHE3 mRNA in the proximal tubule cell is also affected chronically and post-transcriptionally by the PI3K pathway [158,159,160]. It has been demonstrated that insulin also targets Na-K-ATPase, which increases Na reabsorption [161,162]. Insulin enhances Na-K-ATPase activity in the proximal convoluted tubule of rats, as demonstrated by Feraille et al. [163]. The basolateral electrogenic Na-HCO3 cotransporter (NBCe1), which is crucial for sodium and bicarbonate escape from proximal tubular cells, is also known to be stimulated by insulin. As a result, insulin activates all of the transporters needed to absorb Na from the proximal tubules [164]. 

The discovery of insulin receptor substrate (IRS) 1 came about as a result of research into the signal transduction mechanism of insulin [165,166]. However, IRS1/animals survived with only minor IR, which resulted in the discovery of IRS2 [167]. Although IRS1 and IRS2 have very similar structural makeups, their signaling pathways differ [168]. IR and mental impairment also appear in IRS1 and IRS2 knockout mice [169,170]. The tissue expression, IR mechanism, and connection with cell hyperplasia are different between IRS-1 and IRS-2 [171]. Insulin dramatically increased the amount of Na-coupled HCO3 absorption from the proximal tubule in wild-type mice. Insulin’s enhancement of HCO3 absorption was intact in IRS1/mice but dramatically reduced in IRS2/animals. Furthermore, IRS1/mice maintained the insulin-stimulated Akt phosphorylation, which replicated the impact of insulin on proximal absorption, but IRS2/animals drastically reduced it [172]. The tyrosine phosphorylation of IRS2 by insulin was more pronounced than that of IRS1, which is consistent with IRS2 playing a significant role in the insulin-mediated transport stimulation in proximal tubules [173]. Importantly, IR has frequently been linked to IRS1-specific signaling abnormalities [174,175,176,177]. Therefore, hyperinsulinemia may play a key role in the pathophysiology of IR-associated hypertension by facilitating salt retention in an IRS1-independent manner. A schematic diagram showing IR and the possible mechanism for the cause of hypertension is shown in Figure 4. 

The LCHF diet and carbohydrate restriction decreased circulating blood glucose and insulin levels. Hence, the water retention caused by insulin is released and urine volume is increased [178]. When type 2 diabetes (T2D), pre-diabetes, or impaired glucose tolerance (IGT) first emerged in the Norwood primary care (GP) practice in 2013, the patients were recommended to follow a low-carbohydrate diet (defined as 130 g carbs/day) [179]. There were considerable and unanticipated improvements in blood pressure (systolic 148.17 to 133.15 mmHg, *p* = 0.05; and diastolic 91.8 to 83.11 mmHg, *p* = 0.05) in this eight-month pilot trial of 19 patients. This happened despite stopping some antihypertensive medications [179].

## 5. Conclusions 

In T2DM patients, IR is the main cause of worry. In recent years, IR has gained much attention. In the human body, IR is associated with lower insulin sensitivity. Various factors are responsible for IR and these include unhealthy eating habits, lifestyle factors, a lack of physical activity, obesity, dyslipidemia, cardiovascular disease, and other endocrine disorders. In IR, glucose cannot be used from the muscles, fat and liver, and the level of glucose increases in the body. The pancreas may be damaged in the long run which leads to a decrease in insulin production. A LCHF diet may be beneficial, as it reduces glucose and insulin spikes, improves insulin sensitivity, and decreases the chances of developing atherosclerosis. It is important to note that the quantity and quality of carbohydrates and fats may affect glucose and insulin metabolism. Appropriate intake of carbohydrates may also check the glycemic index. IR is associated with hypertension. Consumption of LCHF may result in better insulin response and help control hypertension. 

## Figures and Tables

**Figure 1 biomedicines-11-02271-f001:**
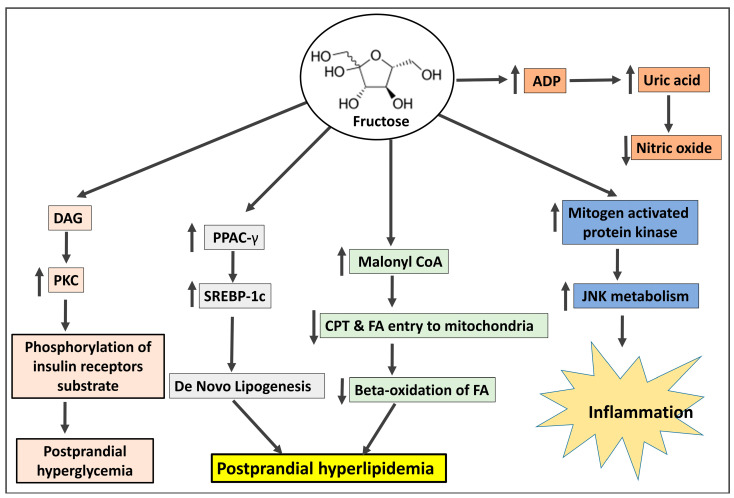
The mechanism by which fructose shares in the pathogenesis of insulin resistance. DAG: diacylglycerol; PKC: protein kinase C; PPAC-γ: peroxisome proliferator-activated receptor γ; SREBP-1c: sterol regulatory element-binding protein 1c; CPT: carnitine-palmitoyl-CoA transferase; FA: fatty acids; ADP: Adenosine di-phosphate; JNK: Jun N-terminal kinase.

**Figure 2 biomedicines-11-02271-f002:**
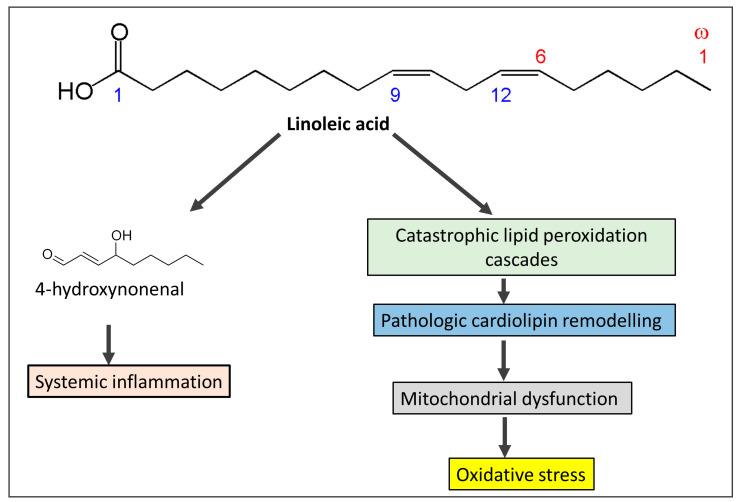
The mechanism by which omega-6 fatty acids derived from seed oils produces systemic inflammation and oxidative stress and share in the pathogenesis of insulin resistance.

**Figure 3 biomedicines-11-02271-f003:**
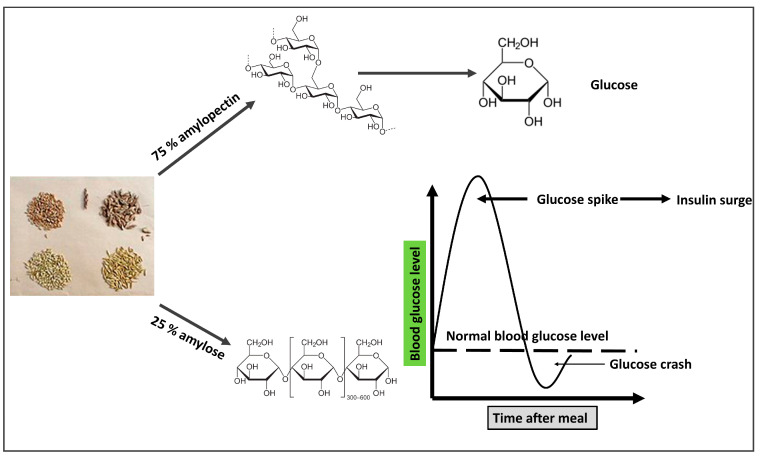
The mechanism by which grains seeds (clockwise from top-left: wheat, spelt, oat, barley) are involved in the pathogenesis of insulin resistance. The branched structure of amylopectin allows more glucose molecules to be hydrolyzed to increase its rate of digestion and increased blood glucose. A straight chain of amylose decreases its digestion rate, so most of it is directed to the large intestine.

**Figure 4 biomedicines-11-02271-f004:**
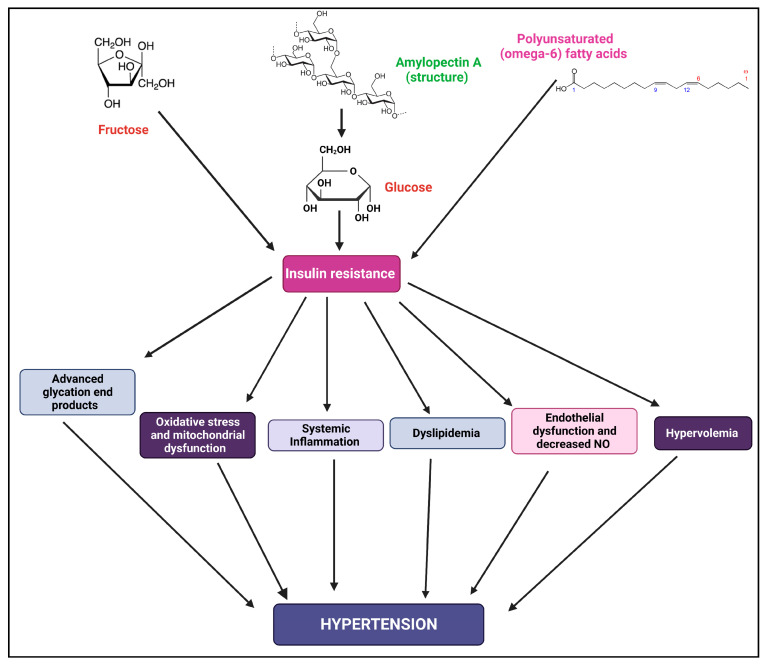
Schematic diagram shows how IR can cause various changes in the body which leads to hypertension.

## Data Availability

Not applicable.

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
