# Peer review of "Insulin Resistance and Hypertension: Mechanisms Involved and Modifying Factors for Effective Glucose Control"

_biomedicines, 2023, doi:10.3390/biomedicines11082271_

Round 1

Reviewer 1 Report

Comments to the Authors of manuscript number: biomedicines-2472022  entitled “Insulin resistance and hypertension: mechanisms involved and modifying factors for effective glucose control”.

The review is well written, but there should be a section on finding the right material, that includes the words used, sides searching etc.

1. L 19 – is an effect of aging on blood vessels not the aging of blood vessels

2. L 20- aging is physiological process but the changes under the influence of improper diet is pathological

3. L 25- should be rephrased

4. L 31 – abbr.

5. L 63, 64- italic

6. L 103- avoid to use “we”

7. L 106- abbr.

8. L 108 – rephrase the sentence, because not from the literature but from clinical observations

9. L 114, 115, 116 – abbr. generally, along the whole text, abbreviation used for the first time should be explained

10. L 119- reference needed

11. L 133 – is it important were? Or the result?

12. L 142, 143 – hormone or hormones?

13. L 151- reference needed

14. L 166 – abbr. should be explained

15. L 178 – abbr.?

16. L 206 – reference needed

17. L 219 – abbr.

18. L 400 – “arterial arteries’?

19. L 449 – IR is used as an abbreviation of insulin resistance. And here?

20. Conclusion is not suitable

Author Response

We appreciate your comments to improve the quality of the article 

Reviewer 2 Report

Thank you for granting me the opportunity to review the article titled "Insulin resistance and hypertension: mechanisms involved and modifying factors for effective glucose control" (biomedicines-2472022), which is submitted to the "Cell Biology and Pathology" section in the Special Issue "Adipose Tissue in Health and Disease."

The aim of this review is to discuss how a low carbohydrate, high fat (LCHF) diet helps maintain normal blood pressure, slows down the progression of atherosclerosis, and reduces the need for medications to control hypertension, also known as the silent killer. Additionally, the review explores the mechanisms associated with insulin resistance (IR) and hypertension, highlighting the potential benefits of LCHF diet in reducing IR and improving glucose control in diabetic patients.

Given the high prevalence of insulin resistance and hypertension in the current population, which is partly attributed to poor lifestyle choices and the aging population resulting from improved health conditions, this topic holds significant importance and will become even more relevant in the future.

Suggestions:

    The title should clearly indicate that it is a review. Reviews play a crucial role in providing an updated overview of a topic within a specific timeframe. They are valuable for updating knowledge and proposing future research.

    The abstract, which is well-presented, should aim to provide a comprehensive overview of the content. In addition to stating the objective of the work, it should include information about the methodology used in the review and highlight the main results, whenever possible. Given the dynamic nature of this field, it would be helpful to mention the specific period covered in the review. This will allow readers to assess the contribution of the work and establish connections with future research.

    It would be beneficial for this review to include a section that assesses the strengths and weaknesses of the current knowledge. By doing so, future research can be proposed to further evaluate the hypotheses presented.

The hypothesis presented suggests that the consumption of a low carbohydrate, high fat (LCHF) diet may be beneficial in reducing insulin resistance and improving glucose control in diabetic patients.

Author Response

(The authors gave the same response as above.)

Reviewer 3 Report

Overall, the paper discusses the relationship between hypertension, insulin resistance, and the effect of different diets, specifically the low carbohydrates and high-fat (LCHF) diet. While the topic is relevant and important, there are several major issues with the paper that warrant its rejection.

Lack of Clarity and Organization:

The paper lacks clarity and organization in presenting the information. The abstract and introduction sections fail to provide a clear outline of the paper's objectives, research questions, and methodology. Additionally, the flow of information is disjointed, making it difficult to follow the main arguments and understand the overall structure of the paper.

Lack of Supporting Evidence:

The paper makes several claims without providing sufficient supporting evidence or references. Statements such as "the LCHF diet is responsible for increasing blood pressure, morbidity, and mortality from hypertension through higher LDL-cholesterol levels" (line 26) and "the LCHF diet maintains blood pressure at its normal levels, slows down atherosclerosis progression" (line 28) need to be supported by relevant studies or data to establish their credibility.

Inadequate Literature Review:

The literature review provided in the paper is insufficient. While some background information is presented, there is a lack of comprehensive analysis and discussion of previous studies related to hypertension, insulin resistance, and the effects of different diets. Including a more thorough review of the existing literature would strengthen the paper's arguments and provide a context for the author's claims.

Lack of Methodology and Data:

The paper does not include any methodology or data to support the claims made. It is essential to provide a clear description of the methodology used to gather and analyze data, as well as present relevant findings to support the arguments made in the paper. Without this information, the claims made in the paper lack scientific rigor and credibility.

Biased Language and Inaccurate Statements:

Throughout the paper, there are instances of biased language and inaccurate statements. For example, the statement that "the LCHF diet reduces glucose and insulin spikes, improves insulin sensitivity, and lessens atherosclerosis risk factors" (line 26) is presented as a factual claim without proper evidence. Additionally, the paper fails to acknowledge potential drawbacks or limitations of the LCHF diet and presents it as a universally beneficial approach.

There too many unjustified auto-citations.

Based on these major issues, I recommend rejecting the paper.

English needs to be improved. The language is often colloquial and unscientific

Author Response

(The authors gave the same response as above.)

Reviewer 4 Report

Authors proposed a paper entitled “Insulin resistance and hypertension: mechanisms involved and modifying factors for effective glucose control” for the publication in Biomedicines, MDPI.

This paper has a quite good scientific soundness and deserves to be published after some minor revisions.

Why the type of this paper is Article if it looks like a review paper?

Here is the list of my issues:

Line 38. Double space among “and” and “a”.

Line 45. “Hypertension has now become a pandemic” are authors sure of this affirmation?

Line 66 “exacerbates IR [8]. IR in” please check it out.

Line 72. Remove the double space here.

Line 85. “with IR [16” double space here

Line 86. “familial” check the spelling.

Improve the focus of Figure 1 and be sure that all the aconyms are defined in the caption and in the manuscript.

Figure 2 also needs to be improved in terms of focus. Moreover, consider the possibility to eliminate it, just describing the mechanism in the manuscript text.

Line 236. “we are going to mention in detail how grains” check syntax and use more impersonal forms.

I suggest not using so much written things in Figure 3. Moreover, I believe that some images could be covered by copyright. Be sure that it is possible to use them.

Line 389. “roles. Environmental” double space here.

A quite good use of English

Author Response

(The authors gave the same response as above.)

Round 2

Reviewer 1 Report

The correction has been done

Reviewer 3 Report

The paper discusses the relationship between hypertension, insulin resistance, and the effect of different diets, specifically the low carbohydrates and high-fat (LCHF) diet. While the topic is relevant and important, there are several major issues with the paper that warrant its rejection.

The paper lacks clarity and organization in presenting the information. The abstract and introduction sections fail to provide a clear outline of the paper's objectives, research questions, and methodology. Additionally, the flow of information is disjointed, making it difficult to follow the main arguments and understand the overall structure of the paper.

Furthermore, the paper makes several claims without providing sufficient supporting evidence or references. Statements such as "the LCHF diet is responsible for increasing blood pressure, morbidity, and mortality from hypertension through higher LDL-cholesterol levels" and "the LCHF diet maintains blood pressure at its normal levels, slows down atherosclerosis progression" need to be supported by relevant studies or data to establish their credibility.

In addition to the lack of supporting evidence, the literature review provided in the paper is insufficient. While some background information is presented, there is a lack of comprehensive analysis and discussion of previous studies related to hypertension, insulin resistance, and the effects of different diets. Including a more thorough review of the existing literature would strengthen the paper's arguments and provide context for the author's claims.

Another significant issue is the absence of a clear methodology and data to support the claims made. It is essential to provide a clear description of the methodology used to gather and analyze data, as well as present relevant findings to support the arguments made in the paper. Without this information, the claims lack scientific rigor and credibility.

Throughout the paper, there are instances of biased language and inaccurate statements. For example, the statement that "Atherosclerosis, or ageing of the blood vessels, is a physiological process accelerated in the last decades by the overconsumption of carbohydrates as the primary sources of caloric intake through increased triglycerides, VLDL-cholesterol, and insulin spikes." is presented as a factual claim without proper evidence. Additionally, the paper fails to acknowledge potential drawbacks or limitations of the LCHF diet and presents it as a universally beneficial approach.

Based on these major issues, I recommend rejecting the paper. The author should address the aforementioned concerns and consider conducting additional research to strengthen their arguments and provide more substantial evidence for their claims.

English needs to be improved